# Large area polymer semiconductor sub-microwire arrays by coaxial focused electrohydrodynamic jet printing for high-performance OFETs

Dazhi Wang [1,2,3] ✉, Liangkun Lu[1], Zhiyuan Zhao[4], Kuipeng Zhao[1], Xiangyu Zhao[1], Changchang Pu[1], Yikang Li[1], Pengfei Xu[1], Xiangji Chen[1], Yunlong Guo [4], Liujia Suo[1], Junsheng Liang[1], Yan Cui[1] & Yunqi Liu [4]

Large area and highly aligned polymer semiconductor sub-microwires were fabricated using the coaxial focused electrohydrodynamic jet printing technology. As indicated by the results, the sub-microwire arrays have smooth morphology, well reproducibility and controllable with a width of ~110 nm. Analysis shows that the molecular chains inside the sub-microwires mainly exhibited edge-on arrangement and the π-stacking direction (010) of the majority of crystals is parallel to the long axis of the sub-microwires. Sub-microwires based organic field effect transistors showed high mobility with an average of 1.9 cm² V⁻¹ s⁻¹, approximately 5 times higher than that of thin film based organic field effect transistors. In addition, the number of sub-microwires can be conveniently controlled by the printing technique, which can subsequently concisely control the performance of organic field effect transistors. This work demonstrates that sub-microwires fabricated by the coaxial focused electrohydrodynamic jet printing technology create an alternative path for the applications of high-performance organic flexible device.

Organic field effect transistors (OFETs) refer to organic polymer semiconductor devices that have a vertical electric field to adjust the lateral channel resistance. OFETs play the roles of amplification and switching in logic gate circuits and sensor units since their channel conductance can be changed by orders of magnitude under the effect of the gate voltage. OFETs are highly suitable for flexible substrates and can draw upon solution method processing techniques, thus allowing these transistors to have broad applications in high-performance flexible sensing, foldable displays[1–4], and wearable devices[5,6]. The performance of OFETs is primarily dependent on the mobility, the on/off ratio, the threshold voltage and the sub-threshold slope. To be specific, the mobility has been found as the most important indicator for assessing the performance of the OFET devices. The mobility can reveal the speed of the carriers moving in the polymer semiconductor layer based on the electric field of OFETs, it is an important vital parameter for assessing the performance of various polymer semiconductor materials[7].

The mobility of the polymer semiconductor layer of OFETs can be controlled by chemical structure, morphology, and molecular orientation. For importanting the mobility of polymer semiconductor, attempts to fabricate aligned organic micro/nano structures by single crystal[8–10], electrospinning[11–13], solution shearing[14,15], photolithography

[1]Laboratory for Micro/Nano Technology and System of Liaoning Province, Dalian University of Technology, Dalian 116024, China. [2]Key Laboratory for Precision and Non-traditional Machining Technology of Ministry of Education, Dalian University of Technology, Dalian 116024, China. [3]Ningbo Institute of Dalian University of Technology, Ningbo 315000, China. [4]Beijing National Laboratory for Molecular Sciences, Key Laboratory of Organic Solids, Institute of Chemistry, Chinese Academy of Sciences, Beijing 100190, China. ✉e-mail: d.wang@dlut.edu.cn

template technology[16–19] have been reported. However, organic micro/nano structures through aforementioned alignment techniques have generally focused on one direction at random placements. Recent research has demonstrated that micro/nano structures are capable of significantly improving the orientation of polymer semiconductor crystals, which can significantly contribute to the anisotropic charge transfer than the thin film structure. Moreover, the performance of the OFETs can be controlled by regulating the number of micro/nanowires[13,20,21]. In order to prepare microwires with high resolution and low cost, Mahalingam designed a pressurized gyration system to realize the mass production of polymer fibers with a width of ~0.8 μm[22,23]. Alenezi further developed the pressurized gyration technique and realized the preparation of core–sheath polymer microwires with a width of ~0.5 μm[24,25]. Wang fabricated wires having a width of ~0.6 μm and a length of 5–20 μm through solvent evaporation, the mobility was found to be more than 2 times of the film[26]. Wu constructed a three-phase assembly system and fabricated one-dimensional wire adopting a capillary-bridge self-assembly strategy, the results show that the mobility of microwires was 2 times than that of the film[27]. Notably, the mentioned fabrication process for microwires is only related to the preparation of minority OFETs with low resolution and less alignment, which is difficult to achieve controllable and large area device preparation, which becomes a significant bottleneck in their further device application. Therefore, the manufacturing technology suitable for large area, controllable and highly aligned sub-microwires based OFETs still remains to be developed[19,21,28].

In this work, a coaxial focused electrohydrodynamic jet (CFEJ) printing technology was developed for the preparation of large area and highly aligned polymer semiconductor sub-microwire arrays. Sub-microwire arrays with a width of 110 nm and 90 nm with a length over 50 mm was produced by the CFEJ printing technology. Two-dimensional grazing-incidence wide-angle X-ray scattering (2D-GIWAXS) analysis shows that the molecular chains inside the PDVT-10 sub-microwires are mainly exhibited edge-on arrangement and parallel to the long axis of the aligned sub-microwires, the π-stacking direction of majority crystalline is orthogonal to the long axis of the aligned sub-microwires with a π-π stacking distance ($d_{π-π}$) of 3.63 Å, which can effectively improve the hole mobility of the OFETs. Sub-microwires based OFETs showed excellent hole mobility in air, with an average mobility of 1.9 cm$^2$ V$^{-1}$ s$^{-1}$, approximately 5 times higher than that of thin film based OFETs (0.4 cm$^2$V$^{-1}$s$^{-1}$). It also exhibited high on/off ratio, which was $1.8 \times 10^5$ with a low threshold voltage of 2.07 V. Moreover, the effect of sub-microwire arrays on the drain current value of the OFETs was investigated. It was observed that the maximum drain current increased proportionally with the increase in the number of sub-microwire arrays. Therefore, the performance of OFETs can be conveniently controlled by the printing of the number of sub-microwires.

## Results

### Characterization of PDVT-10 and silicone oil

Poly [2,5-bis (alkyl) pyrrolo [3,4-c] pyrrole-1,4 (2H,5H)-dione-alt-5,5′-di(thiophen-2-yl)-2,2′-(E)-2-(2-(thiophen-2-yl) vinyl) thiophene] (PDVT-10) is a P-type polymer semiconductor with air stability, which is a typical material used in OFET. The PDVT-10 used in this paper is synthesized in laboratory, its average molecular weights is 47.5 kDa (Supplementary Fig. 1a–c). Another material used in this study was silicone oil, which is used as the outer liquid of CPEJ printing, the viscosity of silicone oil in this work is 60,000 cst, because of the high viscosity it has a larger contact angle than PDVT-10 ink, which can limit the spread of the PDVT-10 wire structurs on the substrate, thereby producing a high printing resolution (Supplementary Fig. 1d, e). More details about the experimental setup are discussed in the Method section.

## Experiment set-up and printing process

Figure 1a illustrates the CFEJ printing system, which consisted of X-Y-Z movement stage, a power supply, two syringe pumps, as well as electrohydrodynamic coaxial needle. The coaxial needle was linked with a high voltage power supply, the inlets were linked with two syringe pumps, in which syringe pump 1 and syringe pump 2 were linked to inner nozzle and external nozzle, respectively. The high viscosity silicon oil and PDVT-10 ink were delivered on the basis of the outer needle and the inner needle, respectively. The high voltage power was used for providing an electric field between the ground electrode and the coaxial needle. Moreover, the syringe pumps were adopted for exerting the hydrodynamic force to promote the ink/solution to the needle outlet.

The CFEJ printing technology can efficiently implement high-resolution patterning of polymer semiconductors on silicon wafer and flexible substrates, which is not achieved by conventional direct writing technology. Specially in this work, a liquid electrode was employed to achieve stable printing process, which is used to neutralize the residual charge and avoid coulomb repulsion phenomena caused by the oncoming printing material and the accumulated residual printed material, the liquid electrode is shown in Supplementary Fig. 2. The liquid electrode device was located below the coaxial nozzle, the substrate controlled by the X–Y movement stage was located between the coaxial nozzle and liquid electrode. Solvent of the isopropanol was filled in the liquid electrode tank. The liquid electrode was heated to 40 °C by a heating plate at the bottom of the electrode tank. During the printing process, the electric field was formed between the coaxial nozzle and the liquid electrode, when the coaxial jet formed and reached the liquid electrode the outer wrapped silicon oil/PDVT-10 solution is rapidly dissolved, which can avoid the accumulation of charge and coulomb repulsion by the oncoming and residual silicon oil/PDVT-10 solution, subsequently eliminates the jet whipping problem usually existed in the common E-Jet printing technique, the printing process is presented in the Supplementary Movie 1. This improves the stability of the coaxial jet and support the consistency of the printed sub-microwire array structures.

Impacted by the high insulation of the outer silicone oil, the surface charge arising from the applied voltage only existed in the interface between the silicone oil and the air, as well as between the silicone oil and the internal PDVT-10 ink (Fig. 1c, d). The movement of surface charge on the inner PDVT-10 ink and the outer silicone oil exterior surface would generate an electrical shearing force on both the inner PDVT-10 ink and the silicone oil. The internal electric shearing force was applied directly to the internal PDVT-10 ink. When the external electric shearing force was applied to the external high viscosity solution, the flow of high viscosity silicone oil could be focused, and then the shearing force would produce high viscosity shearing force and internal pressure. This high viscosity shearing force was also applied to the internal PDVT-10 ink through the liquid-liquid interface. When the electric shearing force, the electric field-induced high viscosity shearing force and the internal pressure were both applied on the internal PDVT-10 ink together, the size of the internal PDVT-10 ink could be significantly reduced and maintained stable at nanoscale. Moreover, the external high-viscosity silicone oil could be used to protect the internal PDVT-10 ink from micro environmental disturbances (e.g., temperature, vibration, as well as airflow).[7,29]

When the electric field force, surface tension, gravity, viscous force and Coulomb force reach equilibrium, the coaxial jet forms and moves toward the liquid electrode, and the strength of the electric field at the outlet of the coaxial needle is expressed as[30]

$$E = 2V/r_c \ln(4H/r_c) \qquad (1)$$

where $H$ is the coaxial needle–electrode spacing, $r_c$ is the inner diameter of the needle, and $V$ is the supply voltage. Moreover, the

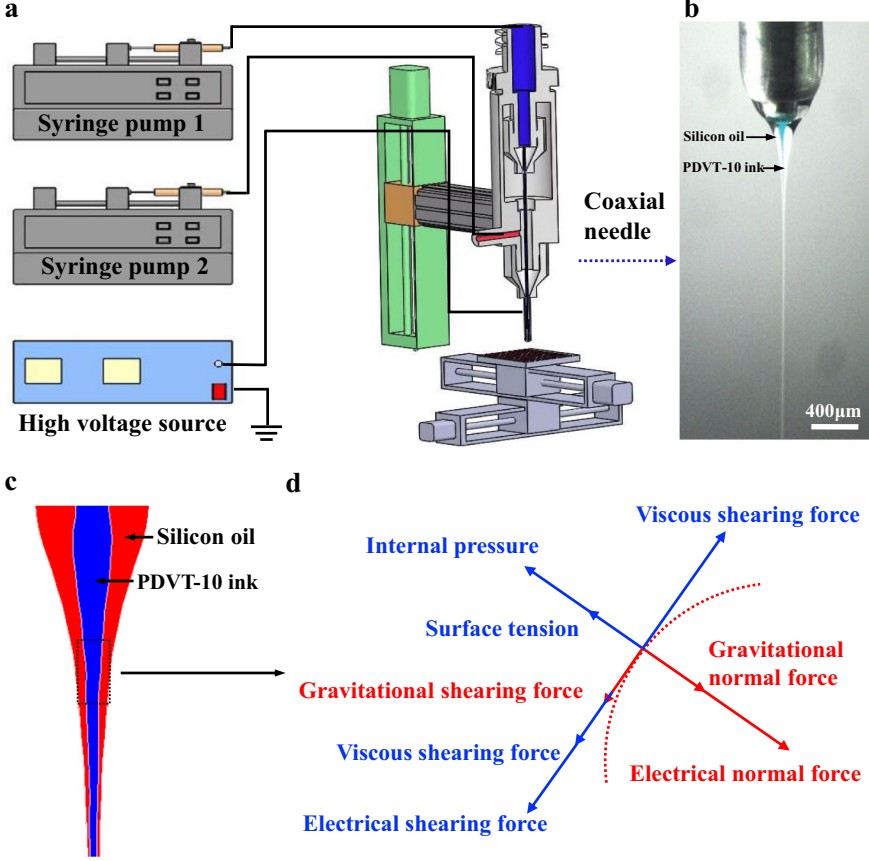

**Fig. 1 | Schematic of the CFEJ printing system and process. a** Schematic of the printing equipment and the coaxial nozzle. **b** Morphology of the coaxial jet during the printing process. **c**, **d** Schematic of the coaxial jet and forces acting on the coaxial jet, where the blue part is the internal PDVT-10 ink and the red part is the external silicon oil.

parameters of the electro fluidic jet have the relationships[31]

$$U = \frac{Q}{d^2}; \frac{\rho Q^2}{d^4} = \varepsilon_0 E_n^2; \frac{\sigma K d^2}{\varepsilon_0} = \frac{\rho Q^3}{d^4}; K d^2 E_s = \frac{\varepsilon_0 E_n Q}{d} \quad (2)$$

where $U$ is the flow rate of the PDVT-10 ink and $Q$ is the inner liquid volume. The Navier–Stokes equations express the conservation of fluid momentum in Fig. 1d as[32]

$$\rho\left(\frac{\partial \vec{u}}{\partial t} + \vec{u} \cdot \nabla \vec{u}\right) = -\nabla p + \vec{F}_e + \vec{F}_S + \vec{F}_V + \vec{F}_g + \vec{F}_i \quad (3)$$

$$Fe = q_v E - \frac{1}{2}E^2\nabla\varepsilon \quad (4)$$

where $F_e$, $F_s$, $F_\mu$, $F_g$, and $F_i$ are the electrical force, surface tension, viscous force, gravitational force and internal pressure, respectively. The relationship between the jet diameter and individual factors is thus expressed as

$$d = Q^{\frac{1}{2}}\left[\frac{F_\mu}{\rho Q} + \left(\frac{2q_v V}{\rho Q r_c \ln(\frac{4H}{r_c})}\right) + g\right]^{-\frac{1}{2}} \quad (5)$$

The above equation shows that the printing structure width is inversely proportional to the viscous shearing force of the silicone oil, the printing speed, and the applied voltage. In this situation, the increase in viscous shearing force and electric field forces lead to an increase in the internal pressure on the polymer solution, thus causing a decrease in the diameter of the inner layer jet. When the printing speed of substrate increased, the lower accumulation of the polymer ink deposited on the substrate, which leads to a smaller size of the printed structure. The printing structure width is proportional to the printing distance and, the polymer ink flow rate. In this situation, decreasing the print distance can reduce the printing width. Additionally, when the polymer flow rate increases, the more accumulation of the polymer ink deposited on the substrate, which leads to a larger size of the printed structure.

Based on suitable working conditions, the stable coaxial jet can be formed, PDVT-10 ink/silicone oil double-layer structure was patterned on silicon wafer (Supplementary Fig. 3a). At this time, the double-layer array structures are needed to be heated at 90 °C for 6 h to reduce the viscosity of the silicone oil, so the polymer will be solidified and stayed on the surface of substrate. Subsequently, the outer silicone oil was removed by isopropanol solution and leaving only PDVT-10 ink, as presented in Supplementary Fig. 3b. Furthermore, the silicon wafer with PDVT-10 polymer sub-microwire arrays were placed on a heating plate to vaporize the remaining solvent (Supplementary Fig. 3c, d).

**Large-area polymer semiconductor sub-microwire arrays by CFEJ printing**

Large-area and highly ordered sub-microwire array structures has been found as the basis for realizing the application of high-performance organic electronic products[27]. In addition, the sinuous structures can substantially increase the tensile strain of flexible devices and improve the stretchability of electronic products. In this work, wafer sized

straight and sinuous sub-microwire array structures were directly printed using the CFEJ printing technology. The concentration of the PDVT-10 ink was 8 mg mL$^{-1}$, and the printing parameters of applied voltage, inner flow rate, outer flow rate, printing distance and printing speed were 4.0 kV, 100 nL min$^{-1}$, 3 μL min$^{-1}$, 3 mm and 400 mm s$^{-1}$, respectively. Subsequently, PDVT-10 ink/silicone oil double-layer structure was prepared using the above method and parameters, as depicted in Fig. 2a, b. The spacing of the PDVT-10 ink/silicone oil structure was 1 mm, thus revealing that the PDVT-10 ink/silicone oil double-layer structure was highly aligned with smooth morphology. Next, the fabricated PDVT-10 ink/silicone oil double-layer structure was processed with the process illustrated in Supplementary Fig. 3 to obtain the sub-microwire arrays containing only PDVT-10 polymer. As depicted in Fig. 2c, the sub-microwire arrays still maintained high straightness and uniformity. Based on the above conditions, the printing distance was increased to 5 mm, and the printing speed was reduced to 200 mm s$^{-1}$, then sinuous structures could be obtained (Fig. 2d, e). The above sinuous structures also exhibit smooth morphology. After the removal of the external silicone oil, the PDVT-10 ink sinuous arrays were obtained (Fig. 2f).

Wearable sensors and flexible displays have been widely demonstrated, their base of flexible substrate should be characterized by scalable and biocompatible. However, the processing of polymer semiconductors on the flexible substrate is still a challenge, and it is extremely important to realize the pattering on insulating flexible substrates[33,34]. In the conventional E-Jet printing process, the charge accumulation and polarization of the insulating substrate can affect the electric field, result in the instability and weak accuracy of the printing results, and even making it difficult to complete the printing process. It is noteworthy that the CFEJ printing technology adopted in this work avoids the accumulation of charge and coulomb repulsion by using the liquid electrode, realizing the stable and high-resolution printing polymer semiconductors on insulating substrates. Moreover, the faster printing speed will also increase the stability of the jet and improve the orderliness of the printing structure. Then sub-microwire arrays on different insulating substrates (PET, PDMS, and PI) was fabricated (Fig. 2g–i). As indicated by the figures, the PDVT-10 ink/silicone oil double-layer structure printed on the insulating flexible substrate still exhibited high parallelism and controllability.

Figure 3 shows various structures produced by CFEJ printing technique of PDVT-10 ink. It was concluded from the analysis (Eq. 5), the printing width can be reduced by decreasing the flow rate, increasing the applied voltage and printing speed. Thus, a lower flow rate (50 nL min$^{-1}$) and printing distance (2 mm), and a higher applied voltage (6 kV) and printing speed (400 mm s$^{-1}$) were employed for printing. It can be seen that the printed PDVT-10 wire arrays has a width of ~110 nm at these working parameters (Fig. 3a). When the flow rate further reduced (~30 nL min$^{-1}$) and other parameters kept constant, the wire width can reach to nanoscale of 90 nm (Fig. 3b). Figure 3c shows the crossed structure of PDVT-10 using CFEJ printing technique. It can be seen that the crossed PDVT-10 sub-microwires contact with each other smoothly, which demonstrates that CFEJ printing can be used to fabricate multilayer and three-dimensional structures. In addition to the typical PDVT-10 polymer ink for high resolution and large area printing, other polymer semiconductor materials including N2200, IDT-BT, and F8BT were also employed for pattering structures with sub-microscale using CFEJ printing technique (Supplementary Fig. 4), which also shown smooth and highly aligned morphologies. This exhibits the wide material application ranges by CFEJ printing technology. Meanwhile, the other remarkable property of sub-microwires is their high ratio of surface area to volume, and the applications of sub-microwires in highly sensitive sensors thus deserves further research.

## Morphology and crystallinity analysis analyses of polymer sub-microwires

The morphology and molecular arrangement of the polymer strongly affects the electronic properties[27,35]. It was reported that the micro/nano scale semiconductor architectures may lead to regular molecular chain alignment and indicate promising field-effect mobility and high-quality OFETs[36,37]. In this work, sub-microwire arrays with smooth morphology was produced by CFEJ printing technology. Furthermore, atomic force microscope (AFM) also shows that the surface of the sub-microwires prepared by CFEJ printing (Supplementary Fig. 5a) appears much smoother behavior than that of the thin film prepared by spin coating (Supplementary Fig. 5b).

2D-GIWAXS is a powerful way to explain the relationship between the molecular packing and the performance of OFETs[36]. Thus 2D-GIWAXS analysis method was conducted for aligned sub-microwire arrays and thin film, the printed sub-microwires and spin coated thin film were all prepared on OTS-modified Si/SiO$_2$ wafers. The incident beam was set orthogonal (I$_o$) (Fig. 4a, e) and parallel (I$_p$) (Fig. 4c) to the length direction of the wires when the GIWAXS analysis was applied on the sub-microwires. It was observed that clear two diffraction signals of (100) and (200) were generated for the PDVT-10 thin films (Fig. 4b), indicating their sufficient crystallinity. In addition, there are (010) diffraction signals in both in-plane and out-of-plane directions, indicating that the thin films have mixed face-on and edge-on stacking inside. It was found that the (010) signal of sub-microwires-I$_p$ only displayed in the in-plane direction (q$_y$ = 1.7–1.8 Å), which indicates that the molecular chains inside the sub-microwires are mainly exhibited edge-on arrangement and parallel to the long axis of the aligned sub-microwires. The π-stacking direction (010) of majority crystalline is orthogonal to the long axis of the aligned sub-microwires with a π-π stacking distance (d$_{π-π}$) of 3.63 Å. The (010) signal is not visible when the incident beam was orthogonal to the long axis of the sub-microwires (Fig. 4f), which indicates that the polymer backbone only aligned perpendicular to the direction of light incidence. The in-plane and out-of-plane 1-D GIWAXS results of the thin films and the sub-microwires are compared and shown in Fig. 4g. It can be seen that the (010) signal only appeared when the incident beam was parallel to the direction of sub-microwires, this confirmed that the molecular chains inside the sub-microwires are mainly exhibited edge-on arrangement and parallel to the long axis of the sub-microwires, which is consistent with the 2-D GIWAXS results above.

Furthermore, high-speed drag behavior is a critical advantage of CFEJ printing technique, which can improve the alignment of molecular chains within polymer. Since the outer silicone oil has high viscosity (6000 cst), the coaxial jet printing onto the substrate with a high speed (~400 mm s$^{-1}$), so the drag force on the sub-microwires is significant. Meanwhile, the direction of the drag force is consistent with the direction of the long axis of the sub-microwires, so this technique promotes the orderly arrangement of polymer molecular chains inside along the long axis.

## Sub-microwire arrays based OFETs

Then sub-microwires based OFETs were designed as shown in Fig. 5a, b. According to the scheme, a batch OFET devices were prepared on the silicon substrate (Fig. 5c). There were 4500 units in total, and the respective unit consisted of gate, dielectric and source/drain electrodes. The silicon wafer comprising 300 nm thick SiO$_2$ as gate and dielectric, parallel source-drain electrodes composed of 30 nm-thick gold were defined on the silicon through photolithography with a channel length of 50 μm. Subsequently, the OTS modification was performed to reduce the defect concentration at the interface. Sub-microwire arrays of PDVT-10 semiconductor were created on source and drain electrodes on the basis of CFEJ printing technique (Fig. 5d, e). As depicted in the figure, the sub-microwire arrays had a compact contact with the electrodes and channels, and the channel of the transistor was clear, without any

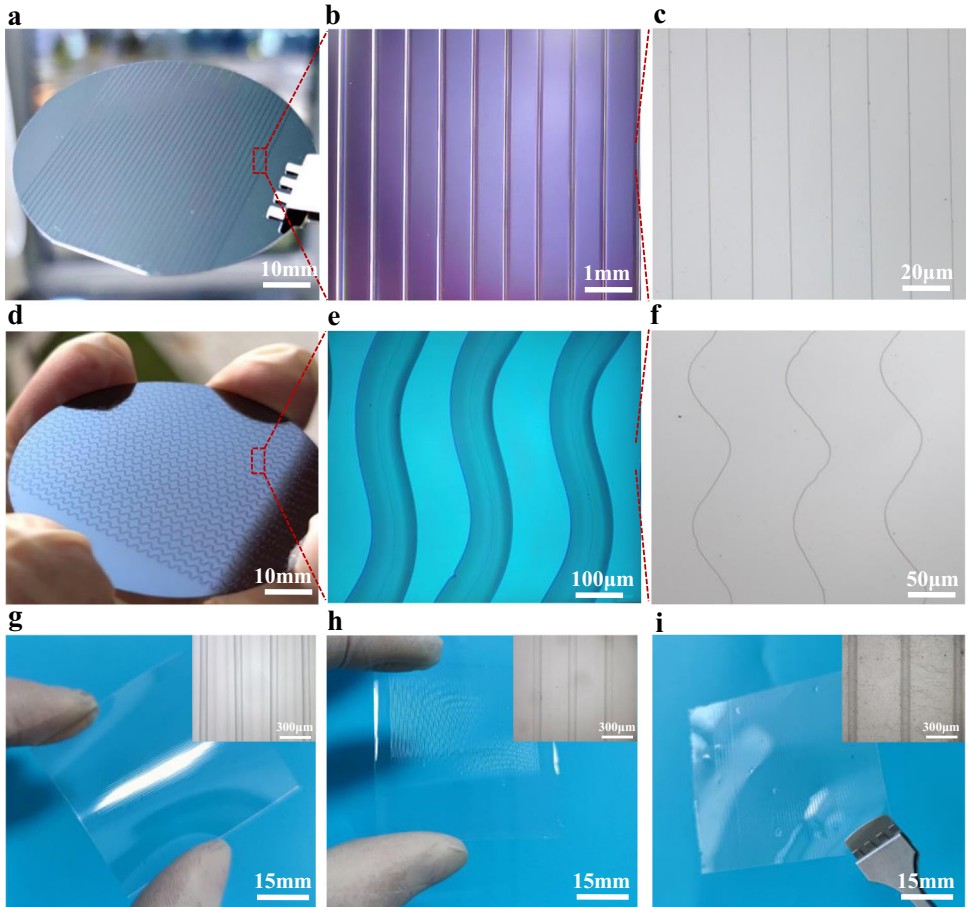

**Fig. 2 | Images of PDVT-10 structures fabricated by CFEJ printing and sub-microwire arrays on different substrates. a–c** Linear array structures printed on silicon wafer. **d–f** Sinuous structures printed on silicon wafer. **g** Linear array structures printed on PET flexible substrate, where the substrate thickness is 200 μm. **h** Linear array structures printed on PDMS substrate, where the substrate thickness is 500 μm. **i** Linear array structures printed on PI substrate, where the substrate thickness is 50 μm.

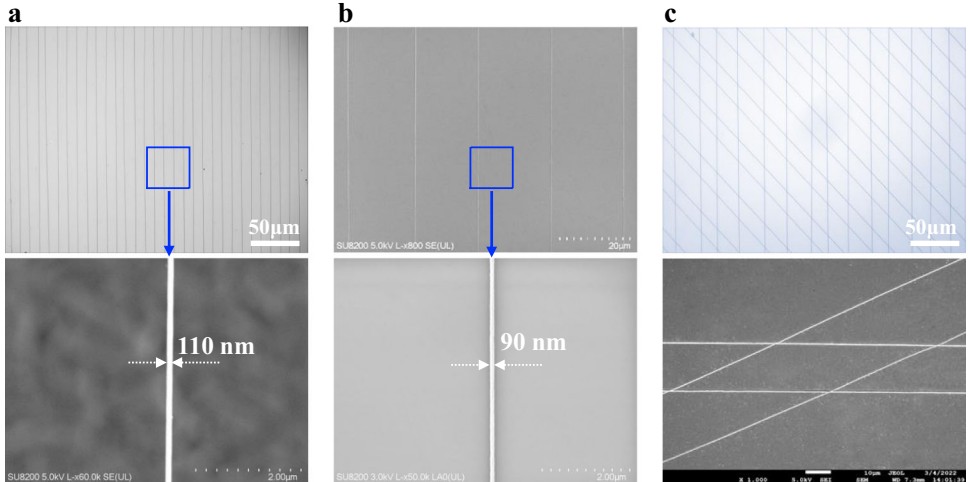

**Fig. 3 | Images of various PDVT-10 micro/nanowire arrays fabricated by CFEJ printing. a** Aligned sub-microwire arrays printed with a width of ~110 nm and its high-magnification image. **b** Aligned nanowire arrays printed with a width of ~90 nm and its high-magnification image. **c** Crossing wire structures printed with a width of ~200 nm and its high-magnification image.

residual polymer impurity. Thus, the CFEJ printing technology opens a way to generate aligned polymer semiconductor materials.

The transfer and output characteristics of single PDVT-10 polymer sub-microwires based OFET are presented in Fig. 5f, g. It can be seen that when the negative gate voltage was applied, a positive carrier hole accumulation layer was formed at the semiconductor sub-microwires and dielectric interface, thus leading to an increase in the current between the source-drain electrodes. Accordingly, the linear region

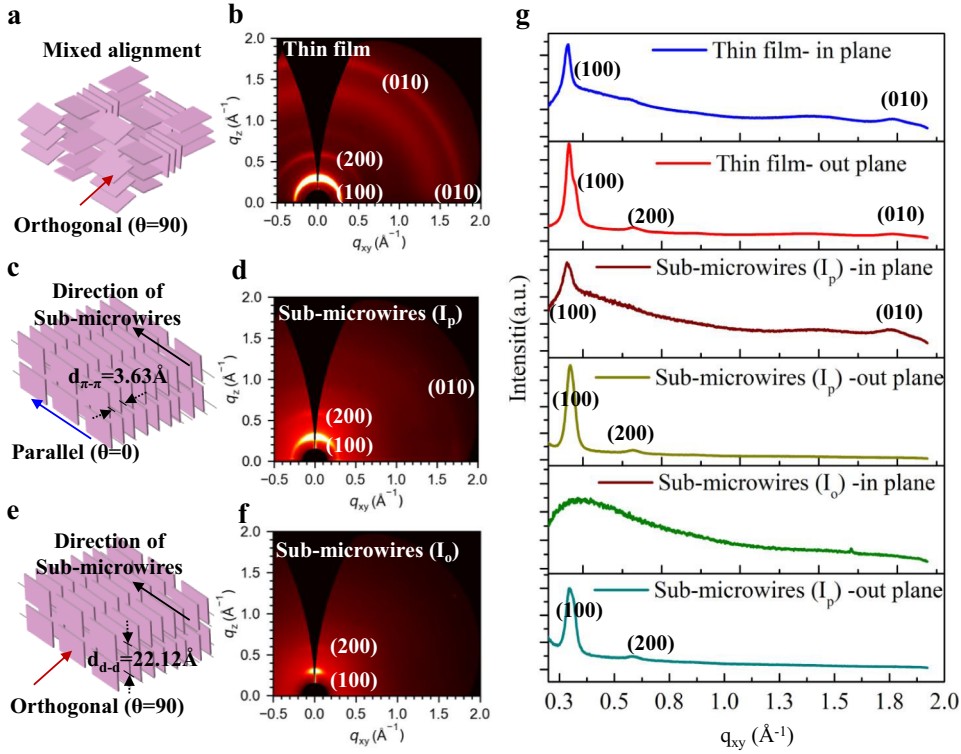

**Fig. 4 | GIWAXS analysis of PDVT-10 sub-microwires and thin film. a** Schematic of the mixed alignment in PDVT-10 films. **b** 2D-GIWAXS analysis of thin film. **c** Schematic of π–π stacking in sub-microwires, the blue arrows show the direction of the incident beam parallel to the long axis of the sub-microwires. **d** 2D-GIWAXS analysis of sub-microwires (incident beam parallel to the length direction of the wires). **e** Schematic of π–π stacking in sub-microwires, the red arrows show the direction of the incident beam orthogonal to the long axis of the sub-microwires. **f** 2D-GIWAXS analysis of sub-microwires (incident beam orthogonal to the length direction of the wires). **g** 1D-GIWAXS results of thin film and sub-microwires.

first appeared in the transfer curve, and then gradually became saturated with the increase in the source voltage. The calculation formula of mobility is as follows: $\mu = 2K^2 L W^{-1} C_i^{-1}$, where $\mu$ is the mobility of OFET, K is obtained by dividing $V_{Gs}$ of the transfer curve by $\sqrt{I_{DS}}$, $L$ is the length of channel, $W$ is the width of the semiconductor layer (width of sub-microwires, if there are multiple wires in the OFET, the width is the sum of all the width of sub-microwires), $C_i$ is the gate dielectric layer capacitance[15,19,37,38]. the values of $L$, $C_i$ are constant as 50 μm and 3.9, the values of $K$, $W$ are calculated as $(1.6 \pm 0.2) \times 10^5$ and 1.1 μm, respectively. The hole mobility was calculated as 1.9 cm² V⁻¹ s⁻¹ (average value of the five OFETs), the mobility of the optimal device was obtained as 3.7 cm² V⁻¹ s⁻¹. The mobility of sub-microwires based OFETs (1.9 cm² V⁻¹ s⁻¹) was 5 times higher than that of thin film based OFETs (0.4 cm² V⁻¹ s⁻¹) at the same conditions (Supplementary Fig. 6). It also exhibited high on/off ratio, which was $1.8 \times 10^5$ with a low threshold voltage of 2.07 V. Furthermore, all OFETs showed low threshold voltage, since the polymer main chain direction was consistent with the sub-microwire direction, and the carrier migration rate along the main chain was significantly higher.

In addition, the influence of the number of sub-microwire arrays in the channel on the drain current value was studied. Different numbers of sub-microwire in the channel of OFETs were prepared, then the transfer curve characteristics was investigated. The results showed that the number of sub-microwires could increase the maximum current value, Fig. 5h showed the dependence of the maximum drain current value on the number of sub-microwires connected to the electrode. Under the condition of $V_{Gs} = -60$ V, the average maximum current values of the OFETs at the voltage range from 0 to −60 V were measured. The current values of OFETs with single sub-microwire number and ten sub-microwire numbers were $0.79 \times 10^{-6}$ A and $4.6 \times 10^{-6}$ A, respectively. When the number of sub-microwire on source-drain electrodes increases to 100, the current value reaches to

$0.998 \times 10^{-4}$ A, which is much higher than that of thin film on the basis of the same conditions, as shown in Fig. 5i.

## Discussion

In summary, the polymer semiconductor sub-microwire arrays structure was fabricated by the CFEJ printing technique. The length of a single sub-microwire was higher than 50 mm, and the width of the wire could be stabilized at ~110 nm. 2D-GIWAXS analysis showed that the molecular chains inside the PDVT-10 sub-microwires mainly exhibited edge on arrangement and the π-stacking direction (010) of the majority of crystals is parallel to the long axis of the aligned sub-microwires. The sub-microwires based OFETs exhibited high hole mobility in air, with the average mobility of 1.9 cm² V⁻¹ s⁻¹ in the saturation regime, five times than the average mobility of thin film based OFETs at the same conditions. Moreover, the on/off ratio was measured as $1.8 \times 10^5$, with a low threshold voltage of 2.07 V. This work demonstrated a method of CFEJ printing for the large area fabrication of polymer semiconductor sub-microwire arrays and high-performance OFETs. In addition, this method also can be used for the fabrication of other micro/nano wire-based devices. Because the CFEJ printing has the great advantages of high resolution and high efficiency, with the implement of automated, assembly and intelligent improvement in the further equipment modification, this method has reliable potential and capability for the realization of micro/nano mass production. Meanwhile, the other remarkable property of sub-microwires is their high ratio of surface area to volume, and the applications of sub-microwires in highly sensitive sensors thus deserves further research.

## Methods
### Materials
In order to making the PDVT-10 ink for CFEJ printing, the PDVT-10 solute was firstly dissolved in anhydrous o-dichlorobenzene solvent,

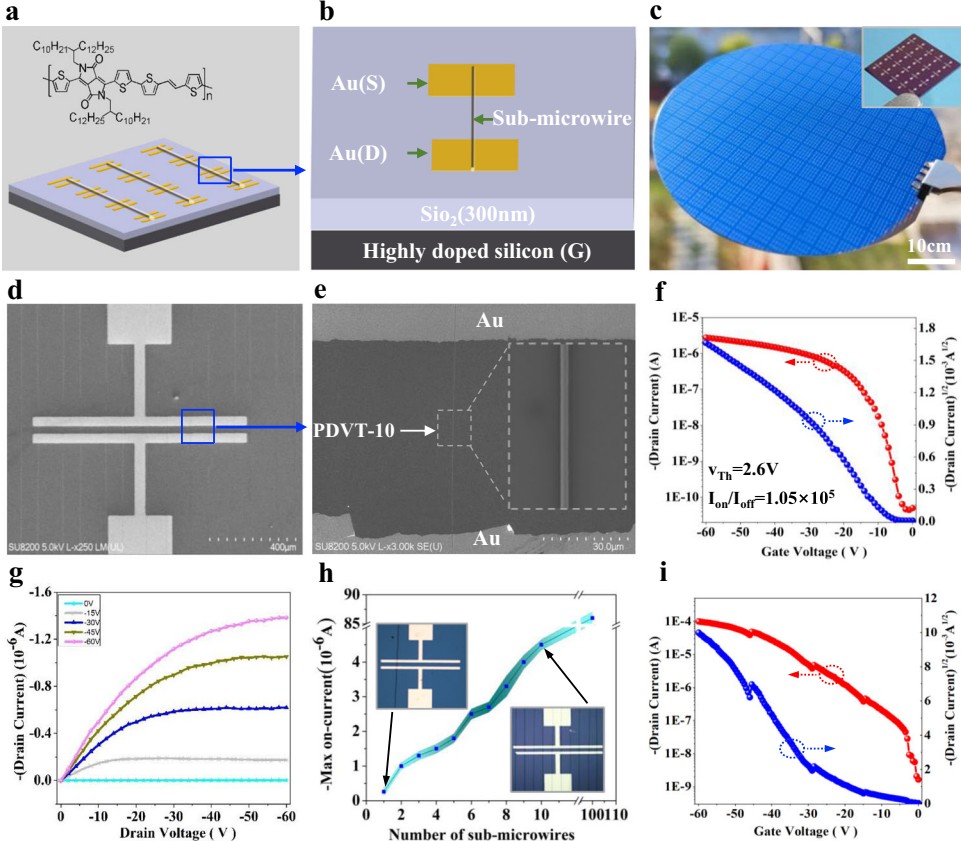

**Fig. 5 | Performance analysis of OFETs. a**, **b** The schematic of the sub-microwires based OFET, the sub-microwire located above the source (S) and drain (D) electrode. **c** Source/drain electrodes with 50 μm channel length, the illustration in the picture shows single parallel electrode. **d**, **e** SEM images of PDVT-10 sub-microwire arrays across two electrodes. **f** Transfer characteristic for the sub-microwires based OFET operated at a constant drain of −60 V. **g** Output curve of sub-microwires based OFET. **h** Effect of the number of sub-microwires on current, the inset shows different numbers of sub-microwires printed on the OFET device. **i** Transfer characteristic for the sub-microwires based OFET operated at a constant drain of −60 V.

and magnetic stirred at a temperature of 90 °C for 60 min, then stirred and heated at 80 °C for 300 min, the fully dissolved PDVT-10 ink is shown in Supplementary Fig. 1c. Before printing, the prepared PDVT-10 ink was filtered through a PTFE filter with a pore size of 0.22 μm to remove undissolved polymer semiconductor. Beside PDVT-10, N2200, IDT-BT and F8BT (Supplementary Fig. 4a, d, g) were also used for CFEJ printing. The solvent for the polymer inks of N2200, IDT-BT and F8BT is also the anhydrous o-dichlorobenzene, the preparation process for these polymer inks was the same as the PDVT-10 ink. Another material used in this study is silicone oil, which is used as the outer liquid of CPEJ printing, the viscosity of silicone oil in this work is 60000 cst.

## Experiment set-up of coaxial focused electrohydrodynamic jet printing

The home-built equipment for CFEJ printing is shown in Fig. 2a. The coaxial needle unit is composed of two stainless steel needles, where precision syringe pump 1 (Harvard Apparatus, Holliston, USA) is connected with inner nozzle (internal diameter 110 μm) the supply rate is 50–100 nL min⁻¹. Syringe pump 2 is connected with external nozzle (internal diameter 460 μm), the supply rate is 3–5 μL min⁻¹. The tip of the coaxial nozzle was connected to a high-voltage power supply (Tianjin Dong Wen, China), its working voltage around 4.2–5.5 kV, the printing distance of the coaxial nozzle to substrate nozzle is 2 mm to collect the sub-microwire arrays from the nozzle, the substrate is connected to the ground, and located on an X-Y-Z three-axis motion platform (HIWIN, Taiwan) controlled by a computer. The liquid electrode device was located below the coaxial nozzle, the substrate controlled by the X-Y movement stage was located between the coaxial

nozzle and liquid electrode. Solvent of the isopropanol was filled in the liquid electrode tank. The liquid electrode was heated to 40 °C by a heating plate at the bottom of the electrode tank. A color camera (Teledyne FLIR, USA) was used to monitor the morphology of the coaxial jet and printed patterns throughout the printing process, in addition, and the coaxial jet was captured by a high-speed cameras (MIKROTRON GmbH CAMCUBE7, Germany). More details about the experimental setup are discussed in the Experimental section.

## Fabrication process and evaluation of OFET

Heavily doped single-sided polished p-type silicon wafer was used as gate, and a layer of SiO₂ oxide (200 nm) was thermally grown by oxidation process. Then the source-drain electrodes of OFET were fabricated on the oxide layer by lithography and sputtering, the channel length between the source-drain electrodes was ~50 μm, the material of the electrodes consists of Cr (5 nm) and Au (30 nm), any of the two electrodes could be used as the test electrode contact. In order to fully remove the photoresist and organic impurities remaining on the silicon wafer. The silicon wafer was firstly immersed in acetone for 120 min, and then immersed in a mixed solution of H₂SO₄: H₂O₂ = 7:3 (volume ratio) for 30 min. Then the silicon wafer was removed and cleaned by deionized water with 20 w power in ultrasonic, followed by deionized water, anhydrous ethanol, acetone cleaning. Finality, the cleaned substrate was dried with nitrogen and treatment with oxygen plasma for 5 min, then placed in a glass culture dish. Then a tiny droplet of OTS was dropped in the middle of the culture dish and heated in a vacuum drying box at 120 °C for 190 min to form an orderly self-assembled monolayer on the surface of SiO₂.

Semiconductor analyzer (Keithley, 4200-SCS, USA) was used to measure the electrical properties of the above OFETs, all devices share a common ground, all tests are conducted in air. The microscopic images were photographed by field emission scanning electron microscope (Hitachi, SU8220, Japan) at 5–15 kV accelerated voltage. The 2D-GIXRD data were obtained at 1W1A, Beijing Synchrotron Radiation Facility, the angle of incidence is 0.2°, the exposure time is 500 s, and the sample-todetector distance (SDD) was 453 mm. The surface morphology wrere photographed by laser confocal microscope (Olympus, OLS4000, apan) and atomic force microscope (Brooke, Nanowizard4XP, Germany).

## Data availability
The associated data used in this study are available in the science date bank database under accession code https://www.scidb.cn/s/aMzEba.

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

## Acknowledgements

This research was supported by National Key R&D Program of China (Grant No. 2018YFA0703200, D.W.), National Natural Science Foundation of China (51975104 and 62074138, D.W. 52003274, Z.Z.), the Fundamental Research Funds for the Central Universities, and Ningbo Institute of Dalian University of Technology. The 2D-GIXRD data were

obtained at 1W1A, Beijing Synchrotron Radiation Facility. The authors gratefully acknowledge the assistance of researchers of the Diffuse X-ray Scattering Station during the experiments. Special thanks are due to Dr. Jidong Zhang in Changchun Institute of Applied Chemistry Chinese Academy of Sciences for his help and advice in the 2D-GIXRD test. We also thank Dr. Huajie Chen in Xiangtan University for sharing PDVT-10 polymer with us and Dr. Cai Rui in Instrumental Analysis Center of Dalian University of Technology for the assistance with AFM analysis.

## Author contributions

D.W. responsible for the funding and resources acquisition, supervising the project, revising the manuscript. L.L. responsible for most of the investigations, methodology development, data collection/analysis, writing and editing the manuscript. Z.Z. helped to take most of the experimental tests and editing the manuscript. K.Z. helped the AFM image. C.P. and X.Z. helped to take the experimental pictures. Y.L., X.C., and P.X. completed the SEM image. L.S., J.L., and Y.C. helped to edit the article. Y.G. and Y.L. helped provide test instruments and test methods. All authors provided comments on the revision of the manuscript.

## Competing interests

The authors declare no competing interests.
