## [Peer Review File · Nature Communications]

Large area polymer semiconductor sub-microwire arrays by coaxial focused electrohydrodynamic jet printing for high-performance OFETsREVIEWER COMMENTS

Reviewer #1 (Remarks to the Author):

In this paper, Wang et al reported large area and highly aligned semiconductor nanowire arrays of PDVT-10 polymer for OFETs by using the coaxial focused electrohydrodynamic jet printing technology. The OFETs-based printed nanowires showed extra high hole mobility with an average of $1.9\text{cm}^2\text{V}^{-1}\text{s}^{-1}$ (maximum of $3.7\text{cm}^2\text{V}^{-1}\text{s}^{-1}$).

Firstly, I think the novelty of this paper is low. The material (PDVT-10) is not new. The manufacturing method (electrohydrodynamic jet printing technology) is as common as the direct writing technology. The performance of the OFET is normal ($1.9\text{cm}^2\text{V}^{-1}\text{s}^{-1}$). The application is missing. I totally do not think this kind of study can be published in Nature Communications.

Second, the picture organization is terrible. Too many pictures (totally 9?) can be moved to the SI.

Third, the mechanism of the improved performance should be discussed in detail. Why the hole mobility of wires is approximately 5 times that of the thin film OFETs? Any experimental evidence?

At the current stage, this work is just at the device level, which lacks scientific depth.

“Nanowire” is confusing, the width of wires is more than 100 nm.

Therefore, the novelty of this paper is hard to satisfy the high-quality requirements of Nature Communications.

Reviewer #2 (Remarks to the Author):

This work by Wang et al. developed a coaxial focused electrohydrodynamic jet printing method for large-area fabrication of highly aligned organic semiconductor nanowires of PDVT-10 polymer. This method produced aligned nanowire arrays with a nanowire diameter of 100 nm. OFETs based on the semiconductor nanowire arrays exhibited good device performance with an average mobility of $1.9\text{cm}^2\text{V}^{-1}\text{s}^{-1}$, high on/off ratio of 1.05×10^5 , and a low threshold voltage of 2.6 V. The mobility was about 5 times higher than that of the film fabricated by conventional methods. This work provides a promising approach to fabricate high-performance organic nanowire array-based OFETs. Therefore, I recommend the acceptance of this manuscript after addressing the following questions.

1. Fig.4b shows the forces acting on the coaxial jet, what is the balance between these forces? How does it influence the printing process? It is better to provide additional explanation.

2. Fig.5 and Fig. 6 show the straight and sinuous nanowire arrays printed by the coaxial focused electrohydrodynamic jet printing. Could the method be used to print crossed nanowires?

3. The authors need to supply the values of each character in the formula, especially the values of conductive channel width.
4. In Fig. 9d, the comparison of mobilities of nanowires and thin films, it is better to provide transfer curves of thin film-based OFETs in Supporting Information.
5. In Fig. 10a, the authors studied the effect of nanowire number in the channel on the device current. I suggest the authors to supply more information about the devices, such as device images with different nanowire number, in the Supporting Information .
6. There are ten figures in this manuscript. I feel it might be too many. Some of the figures could be combined together, such as Figure 9 and Figure 10.

Reviewer #3 (Remarks to the Author):

What are the noteworthy results?

Yes, this is an exciting development.

Will the work be of significance to the field and related fields?

Yes, it will help to scale up the forming of functional materials especially relevant to energy.

How does it compare to the established literature? If the work is not original, please provide relevant references.

Compares very well, a little bit more discussion can be helpful with respect to competing methods: See papers in Appl Phys Rev - Alenezi et al. 92021) and Mahalingam et al. (2020).

Does the work support the conclusions and claims, or is additional evidence needed?

Conclusive. A few sentences on how this innovation will lead to the next stage to boost mass production manufacturing is needed for the readers.

Are there any flaws in the data analysis, interpretation and conclusions? Do these prohibit publication or require revision?

Is the methodology sound? Does the work meet the expected standards in your field?

The SI video can be misleading, what does it show, elaborate. Please discuss this in the main text to eliminate this minor flaw.

Is there enough detail provided in the methods for the work to be reproduced?

Yes.

Comments from Reviewer #1:

Comment 1:

The material (PDVT-10) is not new. The manufacturing method (electrohydrodynamic jet printing technology) is as common as the direct writing technology. The performance of the OFET is normal ($1.9\text{cm}^2\text{V}^{-1}\text{s}^{-1}$). The application is missing.

Answers:

- ✓ In this work, PDVT-10 ink was initially selected as a typical example material for printing. In addition, other organic polymer semiconductors of N2200, TDT-BT and F8BT were used for achieving sub-microwire arrays using coaxial focused electrohydrodynamic jet (CFEJ) printing method (Supplementary Fig. 4) in this version, which also shown smooth and highly aligned morphologies. This exhibits the wide material application ranges by CFEJ printing technology. The mobility of PDVT-10 is not the most outstanding material, however, it is a typical organic semiconductor polymer.
- ✓ The detailed explanation was added in this revision on page 11, paragraph 1, line 204-209: “In addition to the typical PDVT-10 polymer ink for high resolution and large area printing, other polymer semiconductor materials including N2200, IDT-BT and F8BT were also employed for patterning structures with sub-microscale using CFEJ printing technique (Supplementary Fig. 4), which also shown smooth and highly aligned morphologies. This exhibits the wide material application ranges by CFEJ printing technology.”
- ✓ The patterning of polymer semiconductor is great significant for flexible electronic devices. In this study, a CFEJ printing technology was proposed to realize the patterning of large area polymer semiconductor sub-microwire arrays. Compared with the traditional direct

writing methods, the CFEJ printing technology has the noteworthy advantages: (1) high resolution, the printed width is about 90 nm; (2) high speed, the printing speed was about 400 mm s⁻¹. The CFEJ printing technology can efficiently implement high-resolution patterning of polymer semiconductors on silicon wafer and flexible substrates, which is not achieved by conventional direct writing technology. Specially in this work, a liquid electrode was employed to achieve stable printing process, which is used to neutralize the residual charge and avoid coulomb repulsion phenomena caused by the oncoming printing material and the accumulated residual printed material, the liquid electrode is shown in Supplementary Fig. 2. The liquid electrode device was located below the coaxial nozzle, the substrate controlled by the X-Y movement stage was located between the coaxial nozzle and liquid electrode. Solvent of the isopropanol was filled in the liquid electrode tank. The liquid electrode was heated to 40 °C by a heating plate at the bottom of the electrode tank. During the printing process, the electric field was formed between the coaxial nozzle and the liquid electrode, when the coaxial jet formed and reached the liquid electrode the outer wrapped silicon oil/PDVT-10 solution is rapidly dissolved, which can avoid the accumulation of charge and coulomb repulsion by the oncoming and residual silicon oil/PDVT-10 solution, subsequently eliminates the jet whipping problem usually existed in the common E-Jet printing technique.

- ✓ One detailed explanation was added in this revision on page 4, paragraph 3, line 95-109: “The CFEJ printing technology can efficiently implement high-resolution patterning of polymer semiconductors on silicon wafer and flexible substrates, which is not achieved by conventional direct writing technology. Specially in this work, a liquid electrode was employed to achieve stable printing process, which is used to neutralize the residual charge

and avoid coulomb repulsion phenomena caused by the oncoming printing material and the accumulated residual printed material, the liquid electrode is shown in Supplementary Fig. 2. The liquid electrode device was located below the coaxial nozzle, the substrate controlled by the X-Y movement stage was located between the coaxial nozzle and liquid electrode. Solvent of the isopropanol was filled in the liquid electrode tank. The liquid electrode was heated to 40 °C by a heating plate at the bottom of the electrode tank. During the printing process, the electric field was formed between the coaxial nozzle and the liquid electrode, when the coaxial jet formed and reached the liquid electrode the outer wrapped silicon oil/PDVT-10 solution is rapidly dissolved, which can avoid the accumulation of charge and coulomb repulsion by the oncoming and residual silicon oil/PDVT-10 solution, subsequently eliminates the jet whipping problem usually existed in the common E-Jet printing technique, the printing process is presented in the Supplementary movie. This improves the stability of the coaxial jet and support the consistency of the printed sub-microwire array structures.”

- ✓ Another detailed explanation was added in this revision on page 9, paragraph 1, line 177-185: “Wearable sensors and flexible displays have been widely demonstrated, their base of flexible substrate should be characterized by scalable and biocompatible. However, the processing of polymer semiconductors on the flexible substrate is still a challenge, and it is extremely important to realize the patterning on insulating flexible substrates^{33,34}. In the conventional E-Jet printing process, the charge accumulation and polarization of the insulating substrate can affect the electric field, result in the instability and weak accuracy of the printing results, and even making it difficult to complete the printing process. It is noteworthy that the CFEJ printing technology adopted in this work avoids the

accumulation of charge and coulomb repulsion by using the liquid electrode, realizing the stable and high-resolution printing polymer semiconductors on insulating substrates.’’

- ✓ This work mainly focuses on the improvement of mobility from thin film to sub-microwire arrays using a typical material and CFEJ printing technology. The results show that the mobility of sub-microwires based OFETs has approximately 5 times than that of the thin film based OFETs, which is mainly due to the increased molecular arrangement of the sub-microwires compared with that of the thin film, which exhibits great potential in OFETs application. Meanwhile, additional remarkable characteristic of sub-microwires is their high ratio of surface area to volume, which can provide highly sensitive sensor application using sub-microwires based OFETs.
- ✓ One detailed explanation was added in this revision on page 10, paragraph 1, line 196-210: ‘‘Fig. 3 shows various structures produced by CFEJ printing technique of PDVT-10 ink. It was concluded from the analysis (Equation 5), the printing width can be reduced by decreasing the flow rate, increasing the applied voltage and printing speed. Thus, a lower flow rate (50 nL min^{-1}) and printing distance (2 mm), and a higher applied voltage (6 kV) and printing speed (400 mm s^{-1}) were employed for printing. It can be seen that the printed PDVT-10 wire arrays has a width of $\sim 110 \text{ nm}$ at these working parameters (Fig. 3a). When the flow rate further reduced ($\sim 30 \text{ nL min}^{-1}$) and other parameters kept constant, the wire width can reach to nanoscale of 90 nm (Fig. 3b). Fig. 3c shows the crossed structure of PDVT-10 using CFEJ printing technique. It can be seen that the crossed PDVT-10 sub-microwires contact with each other smoothly, which demonstrates that CFEJ printing can be used to fabricate multilayer and three-dimensional structures. In addition to the typical PDVT-10 polymer ink for high resolution and large area printing, other polymer

semiconductor materials including N2200, IDT-BT and F8BT were also employed for patterning structures with sub-microscale using CFEJ printing technique (Supplementary Fig. 4), which also shown smooth and highly aligned morphologies. This exhibits the wide material application ranges by CFEJ printing technology”

- ✓ Another detailed explanation was added in this revision on page 14, paragraph 2, line 281-285: “The hole mobility was calculated as $1.9 \text{ cm}^2 \text{ V}^{-1} \text{ s}^{-1}$ (average value of the five OFETs), the mobility of the optimal device was obtained as $3.7 \text{ cm}^2 \text{ V}^{-1} \text{ s}^{-1}$. The mobility of sub-microwires based OFETs ($1.9 \text{ cm}^2 \text{ V}^{-1} \text{ s}^{-1}$) was 5 times higher than that of thin film based OFETs ($0.4 \text{ cm}^2 \text{ V}^{-1} \text{ s}^{-1}$) at the same conditions (Supplementary Fig. 6). It also exhibited high on/off ratio, which was 1.8×10^5 with a low threshold voltage of 2.07 V.”

Supplementary Fig. 2. **a** Image of CFEJ printing process with solid electrode. **b** Schematic of liquid electrode for CFEJ printing. **c** Image of CFEJ printing process with liquid electrode.

Supplementary Fig. 4. **a-c** Linear array structures of N2200 polymer. **d-f** Linear array structures of IDT-BT polymer. **g-h** Crossing structures of F8BT polymer.

Comment 2:

The picture organization is terrible. Too many pictures (totally 9?) can be moved to the SI.

Answer:

- ✓ We agree with reviewer's suggestion. We have modified and rearranged the figures in this version. The total figures are 5 in the main text, the additional explanations of 6 figures are moved to SI. We believed that the arrangement of the figures is much improved and can satisfy the journal's format and requirement.

Comment 3:

The mechanism of the improved performance should be discussed in detail. Why the hole mobility of wires is approximately 5 times that of the thin film OFETs? Any experimental evidence? At the current stage, this work is just at the device level, which lacks scientific depth.

Answer:

- ✓ We agree with reviewer's suggestion. It was observed in this work that the mobility of sub-microwires based OFETs produce by CFEJ printing is approximately 5 times higher than that thin film based OFETs produced by spin-coating at the same conditions (Supplementary Fig. 6). In order to investigate the scientific mechanism, more experiments and discussions were made in this version. The molecular packing analysis for the sub-microwires and thin film was employed by Two-dimensional grazing-incidence wide-angle X-ray scattering (2D-GIWAXS) method. 2D-GIWAXS is a powerful way to explain the relationship between the molecular packing and the performance of OFETs, thus the 2D-GIWAXS was utilized to analyze the printed sub-microwires and spin-coated films. The results show that the molecular chains in the PDVT-10 sub-microwires are highly arranged in the form of edge-on and parallel to the

long axis of the aligned sub-microwires. The π -stacking direction of majority crystalline is orthogonal to the long axis of the aligned sub-microwires. However, the thin film exhibited mixed stacking of face-on and edge-on. This is the main reason for the high mobility of sub-microwire based OFETs compared with than of the thin film based OFETs.

- ✓ One of the detailed explanations were added in this revision on page 3, paragraph 2, line 64-68: “Two-dimensional grazing-incidence wide-angle X-ray scattering (2D-GIWAXS) analysis shows that the molecular chains inside the PDVT-10 sub-microwires are mainly exhibited edge-on arrangement and parallel to the long axis of the aligned sub-microwires, the π -stacking direction of majority crystalline is orthogonal to the long axis of the aligned sub-microwires with a π - π stacking distance ($d_{\pi-\pi}$) of 3.63 Å, which can effectively improve the hole mobility of the OFETs.”
- ✓ One of the detailed explanations were added in this revision on page 12, paragraph 1, line 217-224: “**Morphology and crystallinity analysis of polymer sub-microwires:** The morphology and molecular arrangement of the polymer strongly affects the electronic properties^{27,35}. It was reported that the micro/nano scale semiconductor architectures may lead to regular molecular chain alignment and indicate promising field-effect mobility and high-quality OFETs^{36,37}. In this work, sub-microwire arrays with smooth morphology was produced by CFEJ printing technology. Furthermore, atomic force microscope (AFM) also shows that the surface of the sub-microwires prepared by CFEJ printing (Supplementary Fig. 5a) appears much smoother behavior than that of the thin film prepared by spin coating (Supplementary Fig. 5b).”
- ✓ One of the detailed explanations were added in this revision on page 12, paragraph 2, line 225-243: “2D-GIWAXS is a powerful way to explain the relationship between the molecular packing and the performance of OFETs³⁶. Thus 2D-GIWAXS analysis method was conducted

for aligned sub-microwires and thin film, the printed sub-microwire arrays and spin coated thin film were all prepared on OTS-modified Si/SiO₂ wafers. The incident beam was set orthogonal (I_o) (Fig. 4a, e) and parallel (I_p) (Fig. 4c) to the length direction of the wires when the GIWAXS analysis was applied on the sub-microwires. It was observed that clear two diffraction signals of (100) and (200) were generated for the PDVT-10 thin films (Fig. 4b), indicating their sufficient crystallinity. In addition, there are (010) diffraction signals in both in-plane and out-of-plane directions, indicating that the thin films have mixed face-on and edge-on stacking inside. It was found that the (010) signal of sub-microwires-I_p only displayed in the in-plane direction ($q_y=1.7-1.8 \text{ \AA}^{-1}$), which indicates that the molecular chains inside the sub-microwires are mainly exhibited edge-on arrangement and parallel to the long axis of the aligned sub-microwires. The π -stacking direction (010) of majority crystalline is orthogonal to the long axis of the aligned sub-microwires with a π - π stacking distance ($d_{\pi-\pi}$) of 3.63 Å. The (010) signal is not visible when the incident beam was orthogonal to the long axis of the sub-microwires (Fig. 4f), which indicates that the polymer backbone only aligned perpendicular to the direction of light incidence. The in-plane and out-of-plane 1-D GIWAXS results of the thin film and the sub-microwires are compared and shown in Fig. 4g. It can be seen that the (010) signal only appeared when the incident beam was parallel to the direction of sub-microwires, this confirmed that the molecular chains inside the sub-microwires are mainly exhibited edge-on arrangement and parallel to the long axis of the sub-microwires, which is consistent with the 2-D GIWAXS results above.’’

- ✓ One of the detailed explanations were added in this revision on page 13, paragraph 2, line 244-249: “Furthermore, high-speed drag behavior is a critical advantage of CFEJ printing technique, which can improve the alignment of molecular chains within polymer. Since the outer silicone

oil has high viscosity (6000 cst), the coaxial jet printing onto the substrate has a high speed ($\sim 400 \text{ mm s}^{-1}$), so the drag force on the sub-microwires is significant. Meanwhile, the direction of the drag force is consistent with the direction of the long axis of the sub-microwires, so this technique promotes the arrangement of polymer molecular chains orderly along the long axis.”

- ✓ Another detailed explanation was added in this revision on page 14, paragraph 2, line 281-284: “The hole mobility was calculated as $1.9 \text{ cm}^2 \text{ V}^{-1} \text{ s}^{-1}$ (average value of the five OFETs), the mobility of the optimal device was obtained as $3.7 \text{ cm}^2 \text{ V}^{-1} \text{ s}^{-1}$ (the mobility of sub-microwires based OFET ($1.9 \text{ cm}^2 \text{ V}^{-1} \text{ s}^{-1}$) was 5 times higher than that of the thin film based OFET ($0.4 \text{ cm}^2 \text{ V}^{-1} \text{ s}^{-1}$)) on the basis of the same conditions (Supplementary Fig. 6).”

Fig. 4 GIWAXS analysis of PDVT-10 sub-microwires and thin film. **a** Schematic of the mixed alignment in PDVT-10 films. **b** 2D-GIWAXS analysis of thin film. **c** Schematic of π - π stacking in sub-microwires, the blue arrows show the direction of the incident beam parallel to the long axis of the sub-microwires. **d** 2D-GIWAXS analysis of sub-microwires. **e** Schematic of π - π stacking in sub-microwires, the red arrows show the direction of the incident beam orthogonal to the long axis of the sub-microwires. **f** 2D-GIWAXS analysis of sub-microwires. **g** 1D-GIWAXS results of thin film and sub-microwires.

Supplementary Fig. 6. Transfer and Output characteristic for the thin film-based OFETs operated at a constant drain of -60 V.

Comment 4:

“Nanowire” is confusing, the width of wires is more than 100 nm.

Answer:

- ✓ We agree with reviewer’s suggestion. In order to avoid confusing, the “nanowire” was replaced by “sub-microwire” in this version including the title and main text. Actually, the CFEJ printing technique allows the patterning of structures with width of micro/nano scale and different materials. In this work, wire arrays with a width of 110 nm was produced at the working parameters of a flow rate of 50 nL min^{-1} , a printing distance of 2 mm, an applied voltage of 6 kV and a printing speed of 400 mm s^{-1} . When the flow rate further reduced ($\sim 30 \text{ nL min}^{-1}$), the wire width can reach to nanoscale of 90 nm. However, the flow rate below 50 nL min^{-1} is not very stable, so the main wire arrays with a width of 110 nm and their OFETs were demonstrated.
- ✓ The detailed explanations were added in this revision on page 10, paragraph 1, line 196-210: “Fig. 3 shows various structures produced by CFEJ printing technique of PDVT-10 ink. It was concluded from the analysis (Equation 5), the printing width can be reduced by decreasing the flow rate, increasing the applied voltage and printing speed. Thus, a lower flow rate (50 nL min^{-1}) and printing distance (2 mm), and a higher applied voltage (6 kV) and printing speed (400 mm s^{-1}) were employed for printing. It can be seen that the printed PDVT-10 wire arrays has a width of $\sim 110 \text{ nm}$ at these working parameters (Fig. 3a). When the flow rate further reduced ($\sim 30 \text{ nL min}^{-1}$) and other parameters kept constant, the wire width can reach to nanoscale of 90 nm (Fig. 3b). Fig. 3c shows the crossed structure of PDVT-10 using CFEJ printing technique. It can be seen that the crossed PDVT-10 sub-microwires contact with each other smoothly, which demonstrates that CFEJ printing can be used to fabricate multilayer and

three-dimensional structures. In addition to the typical PDVT-10 polymer ink for high resolution and large area printing, other polymer semiconductor materials including N2200, IDT-BT and F8BT were also employed for patterning structures with sub-microscale using CFEJ printing technique (Supplementary Fig. 4), which also shown smooth and highly aligned morphologies. This exhibits the wide material application ranges by CFEJ printing technology.”

Fig. 3 Images of various PDVT-10 micro/nanowire arrays fabricated by CFEJ printing. a Aligned sub-microwire arrays printed with a width of ~110 nm and its high-magnification image. **b** Aligned nanowire arrays printed with a width of ~90 nm and its high-magnification image. **c** Crossing wire structures printed with a width of ~200 nm and its high-magnification image.

Comments from Reviewer #2:

Comment 1:

Fig. 4b shows the forces acting on the coaxial jet, what is the balance between these forces? How does it influence the printing process? It is better to provide additional explanation.

Answer:

- ✓ We agree with reviewer's suggestion. In this version, the electric field strength at the nozzle, the fourth-order symmetry analysis of the electrohydrodynamic and the momentum conservation equation of fluid motion (Navier-Stokes equation) were studied, respectively. The relationship between the internal jet diameter and the working parameters was analysed, subsequently, the influence of the forces and working parameters on the printing process was deeply examined.
- ✓ The detailed explanations were added in this revision on page 6-7, paragraph 1-5, line 128-151: “When the electric field force, surface tension, gravity, viscous force and Coulomb force reach equilibrium, the coaxial jet forms and moves toward the liquid electrode, and the strength of the electric field at the outlet of the coaxial needle is expressed as³⁰:

$$E = 2V/r_c \ln(4H/r_c) \quad (1)$$

where H is the coaxial needle–electrode spacing, r_c is the inner diameter of the needle, and V is the supply voltage. Moreover, the parameters of the electro fluidic jet have the following relationships³¹:

$$U = \frac{Q}{d^2}; \frac{\rho Q^2}{d^4} = \varepsilon_0 E_n^2; \frac{\sigma K d^2}{\varepsilon_0} = \frac{\rho Q^3}{d^4}; K d^2 E_s = \frac{\varepsilon_0 E_n Q}{d} \quad (2)$$

where U is the flow rate of the PDVT-10 ink and Q is the inner liquid volume. The Navier–Stokes equations express the conservation of fluid momentum in Fig. 1d as³²:

$$\rho \left(\frac{\partial \vec{u}}{\partial t} + \vec{u} \cdot \nabla \vec{u} \right) = -\nabla p + \vec{F}_e + \vec{F}_s + \vec{F}_v + \vec{F}_g + \vec{F}_i \quad (3)$$

$$F_e = q_v E - \frac{1}{2} E^2 \nabla \mathcal{E} \quad (4)$$

where F_e , F_s , F_v , F_g and F_i are the electrical force, surface tension, viscous force, gravitational force and internal pressure, respectively. The relationship between the jet diameter and individual factors is thus expressed as:

$$d = Q^{\frac{1}{2}} \left[\left(\frac{F_v}{\rho Q} \right) + \left(\frac{2q_v V}{\rho Q r_c \ln \left(\frac{4H}{r_c} \right)} \right) + g \right]^{-\frac{1}{2}} \quad (5)$$

The above equation shows that the printing structure width is inversely proportional to the viscous shearing force of the silicone oil, the printing speed and the applied voltage. The increase in viscous shearing force and electric field forces lead to an increase in the internal pressure on the polymer solution, thus causing a decrease in the diameter of the inner layer jet. When the printing speed of substrate increased, the lower accumulation of the polymer ink deposited on the substrate, which leads to a smaller size of the printed structure. The printing structure width is proportional to the printing distance and the polymer ink flow rate. Decreasing the print distance can reduce the printing width. Additionally, when the polymer flow rate increases, the more accumulation of the polymer ink deposited on the substrate, which leads to a larger size of the printed structure.’’

Comment 2:

Fig. 5 and Fig. 6 show the straight and sinuous nanowire arrays printed by the coaxial focused electrohydrodynamic jet printing. Could the method be used to print crossed nanowires?

Answer:

- ✓ We agree with reviewer's suggestion. More experiments were conducted and crossed wire structures were fabricated in this version, and the printed crossed wires has even surface topography and smooth contact with each other (Fig. 3 and Supplementary Fig. 4), which demonstrates that CFEJ printing can be used to fabricate multilayer and three-dimensional structures.
- ✓ The detailed explanations were added in this revision on page 10, paragraph 1, line 196-210: “Fig. 3 shows various structures produced by CFEJ printing technique of PDVT-10 ink. It was concluded from the analysis (Equation 5), the printing width can be reduced by decreasing the flow rate, increasing the applied voltage and printing speed. Thus, a lower flow rate (50 nL min^{-1}) and printing distance (2 mm), and a higher applied voltage (6 kV) and printing speed (400 mm s^{-1}) were employed for printing. It can be seen that the printed PDVT-10 wire arrays has a width of $\sim 110 \text{ nm}$ at these working parameters (Fig. 3a). When the flow rate further reduced ($\sim 30 \text{ nL min}^{-1}$) and other parameters kept constant, the wire width can reach to nanoscale of 90 nm (Fig. 3b). Fig. 3c shows the crossed structure of PDVT-10 using CFEJ printing technique. It can be seen that the crossed PDVT-10 sub-microwires contact with each other smoothly, which demonstrates that CFEJ printing can be used to fabricate multilayer and three-dimensional structures. In addition to the typical PDVT-10 polymer ink for high resolution and large area printing, other polymer semiconductor materials including N2200, IDT-BT and F8BT were also employed for patterning structures with sub-microscale using

CFEJ printing technique (Supplementary Fig. 4), which also shown smooth and highly aligned morphologies. This exhibits the wide material application ranges by CFEJ printing technology.”

Fig. 3 Images of various PDVT-10 micro/nanowire arrays fabricated by CFEJ printing. a Aligned sub-microwire arrays printed with a width of ~110 nm and its high-magnification image. **b** Aligned nanowire arrays printed with a width of ~90 nm and its high-magnification image. **c** Crossing wire structures printed with a width of ~200 nm and its high-magnification image.

Comment 3:

The authors need to supply the values of each character in the formula, especially the values of conductive channel width.

Answer:

- ✓ We agree with reviewer’s suggestion. The value of each character in the formula was added in this version. The values of L and C_i are constants as $50 \mu\text{m}$ and 3.9 , the value of K is calculated as $(1.6 \pm 0.2) \times 10^{-5}$. The conductive channel width in this work is the width of sub-microwires, and there are multiple sub-microwires in the OFET, so the width is the sum of all the width of sub-microwires ($10 \times 110 \text{ nm}$).

- ✓ The detailed explanations were added in this revision on page 14, paragraph 2, line 276-281: “The calculation formula of mobility is as follows: $\mu=2K^2 L W^{-1} C_i^{-1}$, where μ is the mobility of OFET, K is obtained by dividing V_{G_s} of the transfer curve by $\sqrt{I_{D_s}}$, L is the length of channel, C_i is the gate dielectric layer capacitance. W is the width of the semiconductor layer (width of sub-microwire, if there are multiple wires in the OFET, the width is the sum of all the width of sub-microwires)^{15, 19, 37, 38}, The values of L and C_i are constants as 50 μm and 3.9, the value of K and W are calculated as $(1.6 \pm 0.2) \times 10^{-5}$ and 1.1 μm , respectively.”

Comment 4:

In Fig. 9d, the comparison of mobilities of nanowires and thin films, it is better to provide transfer curves of thin film-based OFETs in Supporting Information.

Answer:

- ✓ We agree with reviewer’s suggestion. In this version, the transfer curves and output curves of thin film based OFETs is shown in the Supplementary Fig. 6. It is found that the mobility of thin film based OFETs is much lower than that of the sub-microwires based OFETs. It is also found that the molecular chains inside the sub-microwires are mainly exhibited edge-on arrangement and parallel to the long axis wires. The π -stacking direction of majority crystalline is orthogonal to the long axis of the aligned sub-microwires.
- ✓ One of the detailed explanations were added in this revision on page 13, paragraph 1, line 244-249: “Furthermore, high-speed drag behavior is a critical advantage of CFEJ printing technique, which can improve the alignment of molecular chains within polymer. Since the outer silicone oil has high viscosity (6000 cst), the coaxial jet printing onto the substrate has a high speed ($\sim 400 \text{ mm s}^{-1}$), so the drag force on the sub-microwires is significant. Meanwhile,

the direction of the drag force is consistent with the direction of the long axis of the sub-microwires, so this technique promotes the arrangement of polymer molecular chains orderly along the long axis.”

- ✓ Another detailed explanation was added in this revision on page 14, paragraph 2, line 282-285: “The hole mobility was calculated as $1.9 \text{ cm}^2 \text{ V}^{-1} \text{ s}^{-1}$ (average value of the five OFETs), the mobility of the optimal device was obtained as $3.7 \text{ cm}^2 \text{ V}^{-1} \text{ s}^{-1}$. The mobility of sub-microwires based OFETs ($1.9 \text{ cm}^2 \text{ V}^{-1} \text{ s}^{-1}$) was 5 times higher than that of thin film based OFETs ($0.4 \text{ cm}^2 \text{ V}^{-1} \text{ s}^{-1}$) at the same conditions (Supplementary Fig. 6).”

Supplementary Fig. 6. Transfer and Output characteristic for the thin film-based OFETs operated at a constant drain of -60 V.

Comment 5

In Fig. 10a, the authors studied the effect of nanowire number in the channel on the device current. I suggest the authors to supply more information about the devices, such as device images with different nanowire number, in the Supporting Information.

Answer:

- ✓ We agree with reviewer’s suggestion. In this version, different numbers of sub-microwire in the channel of sub-microwires based OFETs were prepared, then the OFETs device image with

different sub-microwire numbers was added in Fig. 5h. In addition, the influence of the number of sub-microwire arrays in the channel on the drain current value was studied.

- ✓ The detailed explanations were added in this revision on page 16, paragraph 1, line 296-305: “In addition, the influence of the number of sub-microwire arrays in the channel on the drain current value was studied. Different numbers of sub-microwire in the channel of OFETs were prepared, then the transfer curve characteristics was investigated. The results showed that the number of sub-microwires could increase the maximum current value, Fig. 5h showed the dependence of the maximum drain current value on the number of sub-microwires connected to the electrode. Under the condition of $V_{GS} = -60$ V, the average maximum current values of the OFETs at the voltage range from 0 to -60 V were measured. The current values of OFETs with single sub-microwire number and ten sub-microwire numbers were 0.79×10^{-6} A and 4.6×10^{-6} A, respectively. When the number of sub-microwire on source-drain electrodes increases to 100, the current value reaches to 0.998×10^{-4} A, which is much higher than that of thin film on the basis of the same conditions, as shown in Fig. 5i.”

Fig. 5 Performance analysis of OFETs. **a, b** The schematic of the sub-microwires based OFET. **c** Source/drain electrodes with 50 μm channel length, the illustration in the picture shows single parallel electrode. **d, e** SEM images of PDVT-10 sub-microwire arrays across two electrodes. **f** Transfer characteristic for the sub-microwires based OFET operated at a constant drain of -60 V. **g** Output curve of sub-microwires based OFET. **h** Effect of the number of sub-microwires on current, the inset shows different numbers of sub-microwires printed on the OFET device. **i** Transfer characteristic for the sub-microwires based OFET operated at a constant drain of -60 V.

Comment 6:

There are ten figures in this manuscript. I feel it might be too many. Some of the figures could be combined together, such as Figure 9 and Figure 10.

Answer:

- ✓ We agree with reviewer's suggestion. We have modified and rearranged the figures in this version. The total figures are 5 in the main text, the additional explanations of 6 figures are moved to SI. We believed that the arrangement of the figures is much improved and can satisfy the journal's format and requirement.

Comments from Reviewer #3:

Comment 1:

A little bit more discussion can be helpful with respect to competing methods: See papers in Appl Phys Rev- Alenezi *et al.* (2021) and Mahalingam *et al.* (2020).

Answer:

- ✓ We agree with reviewer's suggestion. We have read the relevant articles and learned more novel methods to fabricate polymer microfibers. Discussions about these methods were added in the introduction section. The relevant articles were also referred in this version, which are listed in the References 22, 23, 24 and 25.
- ✓ The detailed explanations were added in this revision on page 2, paragraph 2, line 48-55: “In order to prepare microwires with high resolution and low cost, Mahalingam designed a pressurized gyration system to realize the mass production of polymer fibers with a width of $\sim 0.8 \mu\text{m}^{22, 23}$. Alenezi further developed the pressurized gyration technique and realized the preparation of core–sheath polymer microwires with a width of $\sim 0.5 \mu\text{m}^{24, 25}$. Wang fabricated wires having a width of $\sim 0.6 \mu\text{m}$ and a length of 5–20 μm through solvent evaporation, the mobility was found to be more than 2 times of the film²⁶. Wu constructed a three-phase assembly system and fabricated one-dimensional wire adopting a capillary-bridge self-assembly strategy, the results show that the mobility of microwires was 2 times than that of the film²⁷.”
- ✓ The added references are listed as follows:
- ✓ 22 S. Mahalingam, R. Matharu, Homer-Vanniasinkam, S. & Edirisinghe, M. Current methodologies and approaches for the formation of core–sheath polymer fibers for biomedical

applications. *Appl Phys Rev* 7, (2020).

- ✓ 23 Amir, A. *et al.* Graphene nanoplatelets loaded polyurethane and phenolic resin fibres by combination of pressure and gyration. *Composites Science and Technology* 129, 173-182 (2016).
- ✓ 24 Alenezi, H., Cam, M. E. & Edirisinghe, M. Experimental and theoretical investigation of the fluid behavior during polymeric fiber formation with and without pressure. *Applied Physics Reviews* 6 (2019).
- ✓ 25 Alenezi, H., Cam, M. E. & Edirisinghe, M. Core–sheath polymer nanofiber formation by the simultaneous application of rotation and pressure in a novel purpose-designed vessel. *Applied Physics Reviews* 8, (2021).

Comment 2:

A few sentences on how this innovation will lead to the next stage to boost mass production manufacturing is needed for the readers.

Answer:

- ✓ We agree with reviewer’s suggestion. More discussions were described in this version for explaining the potential and strategy of CFEJ printing method for micro/nano mass production manufacturing.
- ✓ The detailed explanations were added in the revision on page 16, paragraph 2, line 314-322: “This work demonstrated a method of CFEJ printing for large area fabrication of polymer semiconductor sub-microwire arrays and high-performance OFETs. In addition, this method also can be used for the fabrication of other micro/nano wire-based devices. Because the CFEJ

printing has the great advantages of high resolution and high efficiency, with the implement of automated, assembly and intelligent improvement in the further equipment modification, this method has reliable potential and capability for the realization of micro/nano mass production.’’

Comment 3:

The SI video can be misleading, what does it show, elaborate. Please discuss this in the main text to eliminate this minor flaw.

Answer:

- ✓ We agree with reviewer’s suggestion. Detailed discussions were described in this version for explaining the fabrication process shown in the video. In addition, the video was modified in this version with adding annotation, which will further help readers understand it clearly.
- ✓ The detailed explanations were added in the revision on page 4, paragraph 3, line 95-109: “The CFEJ printing technology can efficiently implement high-resolution patterning of polymer semiconductors on silicon wafer and flexible substrates, which is not achieved by conventional direct writing technology. Specially in this work, a liquid electrode was employed to achieve stable printing process, which is used to neutralize the residual charge and avoid coulomb repulsion phenomena caused by the oncoming printing material and the accumulated residual printed material, the liquid electrode is shown in Supplementary Fig. 2. The liquid electrode device was located below the coaxial nozzle, the substrate controlled by the X-Y movement stage was located between the coaxial nozzle and liquid electrode. Solvent of the isopropanol was filled in the liquid electrode tank. The liquid electrode was heated to 40 °C by a heating plate at the bottom of the electrode tank. During the printing process, the electric field was formed between the coaxial nozzle and the liquid electrode, when the coaxial jet formed and

reached the liquid electrode the outer wrapped silicon oil/PDVT-10 solution is rapidly dissolved, which can avoid the accumulation of charge and coulomb repulsion by the oncoming and residual silicon oil/PDVT-10 solution, subsequently eliminates the jet whipping problem usually existed in the common E-Jet printing technique, the printing process is presented in the Supplementary movie. This improves the stability of the coaxial jet and support the consistency of the printed sub-microwire array structures.’’

The fabricating process of CFEJP technology

REVIEWERS' COMMENTS

Reviewer #2 (Remarks to the Author):

The authors have satisfactorily addressed the reviewers' questions. I recommend the acceptance of this manuscript as it is.

Reviewer #3 (Remarks to the Author):

Responses and revisions are OK.